# Cutaneous expression of growth-associated protein 43 is not a compelling marker for human nerve regeneration in carpal tunnel syndrome

**Liam Carroll, Oliver Sandy-Hindmarch, Georgios Baskozos, Guan Cheng Zhu, Julia McCarthy, Annina Schmid***

Nuffield Department of Clinical Neurosciences, University of Oxford, Oxford, United Kingdom

* annina.schmid@neuro-research.ch

**Data Availability Statement:** Data availability statement: RNA sequencing data is available from dbGaP ID phs001796.v1.p1. Phenotypic data

## Abstract

Growth-associated protein 43 (GAP-43) has long been used as a marker for nerve regeneration following nerve injury, with numerous in vitro and animal studies showing its upregulation in regenerating neurons. In humans, expression of GAP-43 has predominantly been examined in skin biopsies from patients with peripheral neuropathies; with several studies showing a reduction in GAP-43 immunoreactive cutaneous nerve fibres. However, it remains elusive whether cutaneous GAP-43 is a valid marker for human nerve regeneration. Here, we present a cohort of 22 patients with electrodiagnostically confirmed carpal tunnel syndrome (CTS), used as a model system for focal nerve injury and neural regeneration after decompression surgery. We evaluate GAP-43 immunoreactivity and RNA expression levels in finger skin biopsies taken before and 6 months after surgery, relative to healthy controls. We further classify patients as 'regenerators' or 'non-regenerators' based on post-surgical epidermal re-innervation. We demonstrate that patients with CTS have lower GAP-43 positive intra-epidermal nerve fibre density (IENFD) before surgery than healthy controls. However, this difference disappears when normalising for total IENFD. Of note, we found surgery did not change GAP-43 expression in IENF, with no differences both in patients who were classified as regenerators and non-regenerators. We also did not identify pre-post surgical differences in cutaneous GAP-43 gene expression or associations with regeneration. These findings suggest cutaneous GAP-43 may not be a compelling marker for nerve regeneration in humans.

## Introduction

Growth associated protein 43 (GAP-43) is a neuron growth cone-associated member of the neuromodulin family [1,2]. GAP-43 is a key component of the neural growth cone in mammals and is involved in neuro-development and synaptic function but is also thought to be a key player in nerve regeneration following nerve injury through modulation of actin dynamics [3].

including quantification of skin biopsy staining is available from https://doi.org/10.5287/bodleian: R5XEdR18R.

**Funding:** ABS is supported by a Wellcome Trust Clinical Research Career Development Fellowship (222101/Z/20/Z) and the Medical Research Foundation (MRF-160-0013-ELP-SCHM-C0842). Her research is supported by the National Institute for Health Research (NIHR) Oxford Biomedical Research Centre (BRC). The views expressed are those of the authors and not necessarily those of the NHS, the NIHR or the Department of Health. The Oxford carpal tunnel cohort was supported by an advanced postdoc mobility fellowship from the Swiss National Science Foundation (P00P3-158835 to ABS) and an early career research grant from the International Association for the Study of Pain (to ABS). GB is supported by Diabetes UK (19/0005984). Guan Cheng Zhu was supported by the Global Networking Talent 3.0 Plan from the Ministry of Education, Taiwan (R.O.C.). This research was funded in whole, or in part, by the Wellcome Trust [222101/Z/20/Z]. For the purpose of Open Access, the author has applied a CC BY public copyright license to any Author Accepted Manuscript version arising from this submission.

**Competing interests:** The authors have declared that no competing interests exist.

Multiple pre-clinical studies have investigated the role of GAP-43 in peripheral nerve regeneration: GAP-43 has consistently been shown to be upregulated in regenerating but not non-regenerating DRG neurones [4,5], dorsal horn of the spinal cord [4] and peripheral axons [6]. Expression of a GAP-43 transgene enhances neurite outgrowth in PC12 cells *in vitro* [7]. In addition, overexpression of GAP-43 increases the sprouting of neurites following peripheral nerve injury whereas its suppression results in reduced sprout-promoting activity [8].

In humans, expression of GAP-43 has been predominately examined in skin biopsies. In healthy human skin, GAP-43 is present in intraepidermal and dermal nerve fibres as well as in fibres innervating Meissner corpuscles and sweat glands [9,10]. Several studies have reported a reduction in GAP-43 immunoreactive cutaneous nerve fibres in patients with peripheral neuropathies [11–14]. However, it remains unclear whether this reduction is simply a reflection of the simultaneous decrease in overall cutaneous nerve fibres, as the proportion of GAP-43 + intraepidermal nerve fibres from total nerve fibres remains unchanged or increased in some studies [14–16] whereas it is decreased in other studies [11,12]. Importantly, the validity of GAP-43 in cutaneous neurons as a marker for nerve regeneration has not been conclusively examined in humans. Only few studies have investigated the temporal pattern of GAP-43 expression. Narayanaswamy et al [13] did not observe changes in subepidermal GAP-43 staining over a 6 months period in patients with diabetic polyneuropathy even though the total number of PGP9.5+ axons decreased over time. In a model of topical capsaicin induced nerve degeneration and subsequent regeneration, GAP-43 subepidermal nerve fibres decreased acutely and increased back to normal values on day 26. However, relationships between GAP-43 expression and extent of nerve regeneration were not examined. A recent study reported unchanged GAP-43+ intraepidermal nerve fibre density following chemotherapy, and no association with epidermal innervation [15]. Therefore, the role of GAP-43 as a marker for axon regeneration in human skin needs to be examined.

We have previously used carpal tunnel syndrome (CTS) as a model system to prospectively study human sensory nerve regeneration [17]. Specifically, carpal tunnel decompression surgery allows the longitudinal observation of human sensory nerve regeneration from injury (pre-operatively) to recovery (post-operatively). CTS is characterised by axon degeneration predominantly of the small fibre population [18], which shows signs of regeneration after surgery [17]. CTS is therefore an optimal model system to determine the role of GAP-43 as a marker for sensory axon regeneration following focal nerve injury in humans.

In this study, we compare the expression of GAP-43 protein and gene expression levels in the skin of patients with CTS compared to healthy age and gender matched controls and also determine any differences in GAP-43 expression in patients' skin before and after decompression surgery. We further examine whether GAP-43 expression is related to the regenerative capacity of intraepidermal nerve fibres in patients with CTS.

## Methods

### Patients

This study is a post hoc analysis of data from the Oxford CTS cohort [17]. Patients with clinically and electrodiagnostically confirmed CTS were recruited from the surgical waiting lists at Oxford University Hospitals NHS Foundation Trust. Patients were excluded if electrodiagnostic testing revealed abnormalities other than CTS, if another medical condition affecting the upper limb or neck was present (i.e. hand osteoarthritis, cervical radiculopathy), if there was a history of significant trauma to the upper limb or neck, or if CTS was related to pregnancy or diabetes. Patients undergoing repeat carpal tunnel surgery were excluded. Here, we selected 22 patients who were previously classified as regenerators (n = 11) and non-regenerators (n = 11)

as determined on skin biopsies (see below) [17]. Eleven healthy volunteers (proportionally age- and gender-matched to the CTS surgery group) without any systemic medical condition, no history of hand, arm or neck symptoms, and normal electrodiagnostic testing served as healthy controls. Whereas healthy participants attended only one appointment, patients attended two appointments: one before carpal tunnel decompression and one 6 months after surgery. The study was approved by the national ethics committee (London Riverside Ref 10/H0706/35, Oxford C 18/SC/0263) and was conducted in compliance with the declaration of Helsinki. All participants gave informed written consent before participating in this study. Detailed aspects of the cohort have previously been reported [17].

## Clinical phenotype

Clinical phenotype included demographic (age, gender) as well as clinical data such as symptom duration. Symptom severity and functional deficits were evaluated with the Boston Carpal tunnel questionnaire (containing both symptom and function scales; 0 = no symptoms/disability, 5 = very severe symptoms/disability) [19]. Clinical recovery for symptoms was determined with a global rating of change scale (GROC) [20]. The GROC scale ranges from 'a very great deal better' (+7), to 'a very great deal worse' (-7). A change of $\geq 5$ (a good deal better/worse) is clinically meaningful [21]. We also performed electrodiagnostic tests with an ADVANCE system (Neurometrix, USA) to determine sensory and motor latencies and amplitudes of the median, ulnar and radial nerve. The protocol has been detailed elsewhere [18]. Electrodiagnostic test severity was graded from very mild to extremely severe [22]. Absent sensory and motor recordings were replaced with values of zero for amplitudes but excluded from analysis of latencies and nerve conduction velocities to prevent inflated values.

## Skin biopsies

Skin biopsies (3mm diameter) were taken on the ventrolateral aspect of the proximal phalanx of the index finger, innervated by the median nerve in both healthy participants and patients with CTS pre surgery. In CTS patients, a second biopsy was taken 6 months after surgery. This biopsy was taken a few millimetres more proximal to avoid the primary biopsy site. The biopsies were performed under sterile conditions following local anaesthesia with 1% lidocaine (1–1.8ml). Biopsy samples were fixed in fresh periodate-lysine-paraformaldehyde for 30 minutes before being washed in 0.1M phosphate buffer and cryoprotected in 15% sucrose in 0.1M phosphate buffer. The tissue was embedded in OCT, frozen and stored at -80 degrees. To assure consistency, histological analysis was performed by the same blinded observer.

Serial biopsies pre- and post-surgery of the same patient were processed simultaneously to minimise variability. Fifty micrometer (μm) sections were cut on a cryostat. Sections were blocked for 1 hour at room temperature in a solution of 1% BSA/2% milk powder/0.3% Triton X-100 before incubation with primary antibodies for protein gene product 9.5 (PGP 9.5, Bio-Rad UK, 1863–1004, 1:300) and growth-associated protein 43 (GAP-43, Abcam UK, Ab12274, 1:500) over three nights at 4˚ Celsius. Sections were washed 3x10 minutes in PBS-Tx-100 before incubation in biotin (goat anti mouse, BA9200, 1:200) in the dark for 2 hours at room temperature. Sections were washed again 3x5 minutes in PBS before secondary antibody (CY3, Stratech, 111 165 144, 1:500) and Streptavidin (AlexaFluor 488, Thermofisher UK, S11223, 1:500) for 2 hours at room temperature in the dark. Stained samples were then washed with PBS-Tx prior to coverslip mounting with VECTASHIELD® Mounting Medium with DAPI counterstain.

Intraepidermal nerve fibre density (IENFD) was established for both PGP and GAP-43 stains by counting 3 sections per participant down the microscope (Leica DM 2500

microscope, Leica Germany) following established guidelines [23]. Sections were then photographed at 5x magnification (Leica DM 2500 microscope, Leica Germany) and epidermal length was measured with ImageJ (NIH USA). Average IENFD was expressed as fibres/mm epidermis for both PGP+ and GAP-43+ axons. We additionally report the GAP-43 data as the percentage of PGP fibres containing GAP-43 to correct for differences in PGP+ IENFD.

Patients were classified into regenerators and non-regenerators by comparing their PGP + IENFD post to pre surgery. Patients with an increase in PGP+ IENFD post compared to pre-surgery were classified as regenerators whereas patients with no change or a decrease in IENFD post-surgery were classified as non-regenerators.

## Gene expression data

We further enquired our previously published dataset on gene expression in index finger skin of n = 46 patients with CTS before and 6 months after surgery, which was collected in the same extended patient cohort (dbGaP ID phs001796.v1.p1). Details on the RNA sequencing experiment and analysis can be found elsewhere [17].

## Statistical analyses

Statistical analyses were performed with SPSS Statistics (IBM, Version 27) and R [24]. After inspection of histograms and Shapiro-Wilk testing, nonparametric statistics were used to accommodate non-normality of data. For histological data, the GAP-43+ IENFD and percentages of PGP+ IENF containing GAP-43 were compared between healthy participants and patients before surgery with Mann Whitney U tests and between patients pre and post-surgery with Wilcoxon signed rank tests. To determine a potential effect of differential regenerative capacity, we compared the percentage of PGP+ IENF containing GAP-43 between patients classified as regenerators and non-regenerators both before and after surgery using Mann Whitney U tests. We also compared pre-operative PGP+ IENFD between regenerators and non-regenerators using a Mann Whitney U test to assure potential effects were not due to pre-operative differences in IENFD. Finally, we used Spearman correlation analyses to explore associations between the percentage of PGP+ IENF containing GAP-43 and regeneration capacity (determined as the difference in PGP+ IENF post-pre surgery).

For gene expression analyses, we compared normalised log2 counts of GAP-43 before and after surgery with Wilcoxon signed rank tests. We also performed a subgroup analysis in patients classified as regenerators and non-regenerators, comparing both pre and post operative data with Mann Whitney U tests. We repeated the same analyses for GAP-43 expressed as a percentage of PGP95 gene expression to account for potential differences in overall axon density.

## Results

Demographic and clinical characteristics of the participants can be found in Table 1. After surgery, 19 patients (86%) reported a clinically meaningful improvement in their symptoms according to the GROC scale. Of the 22 patients, 11 were classified as regenerators and 11 as non-regenerators. Their clinical characteristics can be found in Table 2.

## GAP-43 protein expression is not associated with intraepidermal nerve fibre regeneration

GAP-43+ IENFD was reduced in patients with CTS before surgery compared to healthy controls with no difference in GAP-43+ IENFD before and after surgery (median [IQR] healthy

**Table 1. Participant characteristics.**

|  | CTS patients pre | CTS patients post | Healthy controls |
|---|---|---|---|
| Number of participants | 22 | 22 | 11 |
| Gender (female/male) | 15/7 |  | 7/4 |
| Mean Age (SD) [years] | 62.5 (11.2) |  | 62.4 (11.4) |
| Mean weight (SD) [kg] | 70.5 (15.4) |  | 73.6 (14.7) |
| Mean height (SD) [cm] | 167.2 (8.4) |  | 171.2 (8.0) |
| Mean symptom duration (SD) [months] | 37.6 (47.9) |  |  |
| Mean Boston scale (SD) |  |  |  |
| Symptoms | 2.73 (0.70) | 1.55 (0.54) |  |
| Function | 2.21 (0.79) | 1.58 (0.60) |  |
| Global rating of change* |  | 6 [2] |  |
| Electrodiagnostic test severity |  |  |  |
| Normal, n (%) | 0 (0) | 4 (18) |  |
| Very mild, n (%) | 3 (14) | 5 (23) |  |
| Mild, n (%) | 5 (23) | 1 (4.5) |  |
| Moderate, n (%) | 8 (36) | 7 (31.5) |  |
| Severe, n (%) | 1 (4.5) | 3 (14) |  |
| Very severe, n (%) | 4 (18) | 2 (9) |  |
| Extremely severe, n (%) | 1 (4.5) | 0 (0) |  |

*median and interquartile range.

3.5 [5.9] fibres/mm epidermis, CTS pre 0.3 [1.5], CTS post 0.6 [1.1], Mann Whitney U p = 0.021, Wilcoxon p = 0.498, Fig 1A).

However, when correcting for the total number of axons, there was no difference in the percentage of PGP+ IENF containing GAP-43 among groups (median [IQR] healthy 7.96% [23.54%], CTS pre 13.1% [26.9%], CTS post 14.6% [41.2%], Mann Whitney p = 0.872, Wilcoxon p = 0.701, Fig 1B).

Eleven patients each were classified as regenerators and non-regenerators according to the difference in their PGP+ IENFD post to pre surgery (Fig 1C). These groups had comparable

**Table 2. Participant characteristics of patients classified as regenerators and non-regenerators.**

|  | Regenerators | Non-regenerators |
|---|---|---|
| Number of participants | 11 | 11 |
| Gender (female/male) | 8/3 | 7/4 |
| Mean Age (SD) [years] | 61.8 (11.4) | 63.2 (11.5) |
| Mean weight (SD) [kg] | 75.0 (16.7) | 65.9 (13.2) |
| Mean height (SD) [cm] | 167.7 (9.4) | 166.8 (7.7) |
| Mean symptom duration (SD) [months] | 24.4 (10.9) | 50.7 (65.7) |
| Mean Boston scale (SD) |  |  |
| Symptoms | 2.69 (0.65) | 2.77 (0.78) |
| Function | 2.16 (0.88) | 2.25 (0.73) |
| Electrodiagnostic test severity |  |  |
| Normal, n (%) | 0 (0) | 0 (0) |
| Very mild, n (%) | 1 (9.5) | 2 (18) |
| Mild, n (%) | 4 (36) | 1 (9.5) |
| Moderate, n (%) | 3 (27) | 5 (45) |
| Severe, n (%) | 0 (0) | 1 (9.5) |
| Very severe, n (%) | 2 (18) | 2 (18) |
| Extremely severe, n (%) | 1 (9.5) | 0 (0) |

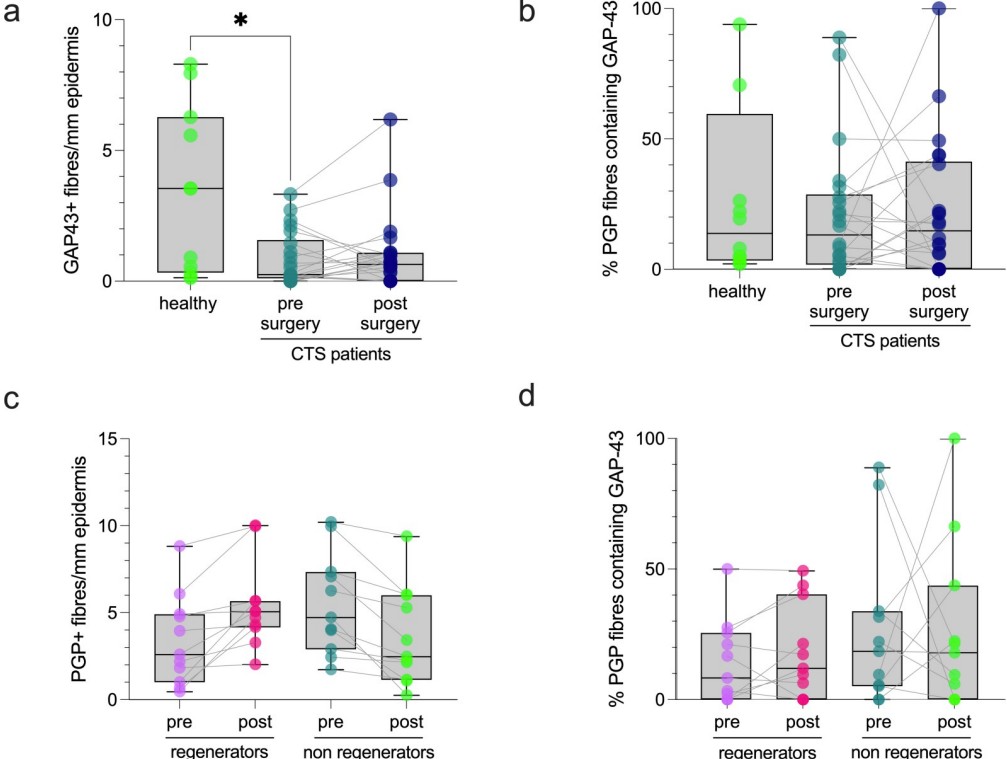

**Fig 1. Immunohistochemical staining results of GAP-43 and PGP.** (A) Patients with CTS before surgery have fewer GAP-43+ IENF than healthy controls. No changes in GAP-43+ IENFD was apparent from before to six months after surgery in patients with CTS. (B) No difference was apparent in the % of PGP+ IENF containing GAP-43 among healthy participants, patients with CTS before and after surgery. (C) IENFD of patients with CTS classified as regenerators and non-regenerators according to the difference in PGP+ IENFD before and after carpal tunnel surgery. (D) No difference was apparent in the percentage of PGP+ IENF containing GAP-43 between regenerators and non-regenerators both pre- and post-surgery. *p = 0.036.

pre-operative PGP+ IENFD (median [IQR] regenerators 2.6 [3.9] fibres/mm epidermis; non-regenerators 4.7 [4.4], Mann Whitney U test p = 0.088). There was no difference in the percentage of PGP+ IENF containing GAP-43 between patients with CTS classified as regenerators and non-regenerators both before (median [IQR] regenerators 8.3% [25.6%]); non-regenerators 18.5% [28.8%], Mann Whitney U test p = 0.332) and after surgery (regenerators 12.0% [40.3%]; non-regenerators 18.0% [43.8%], Mann Whitney U test p = 0.748, Fig 1D).

Correlation analyses demonstrated no significant association between PGP+ IENF regeneration (post-pre-PGP+ IENFD) and the percentage of PGP+ IENF containing GAP-43 both pre- and post-surgery (Spearmans' rho pre surgery -0.08, p = 0.719; post-surgery -0.02, p = 0.948, S1 Fig). An example of PGP and GAP-43 immunostaining can be found in Fig 2.

## GAP-43 gene expression is comparable between regenerative and non-regenerative phenotypes

GAP-43 gene expression levels (median normalised log2 counts CTS pre 4.01 [IQR 0.30], CTS post 4.02 [0.32], Wilcoxon p = 0.983, Fig 3A) and GAP-43 expressed as a percentage of PGP9.5 gene expression (median percentage CTS pre, 47.7 [IQR 3.3] CTS post 47.7 [3.7], Fig 3B, Wilcoxon p = 0.974) were comparable in patients before and after surgery. There were also no differences in GAP-43 gene expression (pre-operative median normalised log2 counts [IQR]

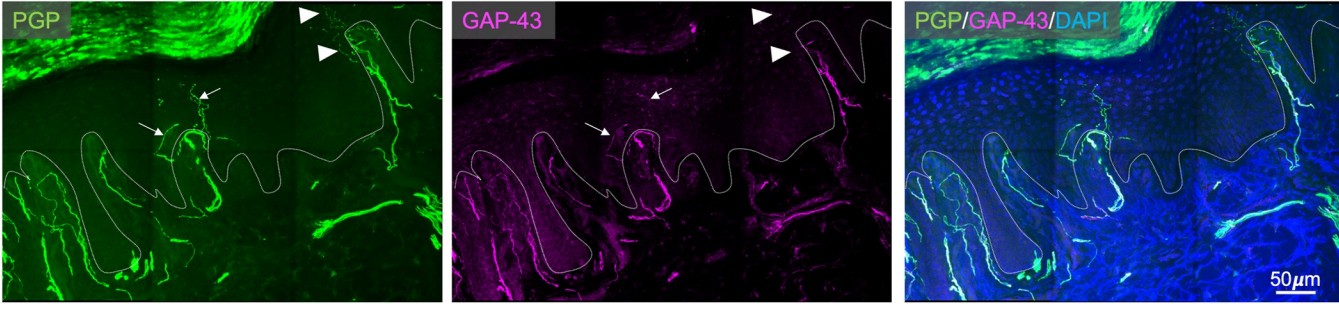

**Fig 2. Example of skin immunostaining for PGP 9.5, GAP-43 and merged staining.** Some PGP9.5+ intraepidermal nerve fibres are double labelled with GAP-43 (arrows) whereas others are not (arrow heads). The dermal-epidermal border is indicated with the dotted line.

regenerators 4.02 [0.27]; non-regenerators 3.98 [0.45], Mann Whitney U test p = 0.736; post-operative regenerators 4.01 [0.28], non-regenerators 4.10 [0.47], Mann Whitney U test p = 0.964, Fig 3C) and GAP-43 expressed as a percentage of PGP9.5 gene expression (pre-operative median percentage [IQR] regenerators 47.6 [3.2], non-regenerators 47.9 [4.6], Mann Whitney U test p = 0.770, Fig 3C; post-operative regenerators 47.7 [2.6], non-regenerators 48.0 [5.8], Mann Whitney U test p = 0.946, Fig 3D) between patients classified as regenerators and non-regenerators both before and after surgery.

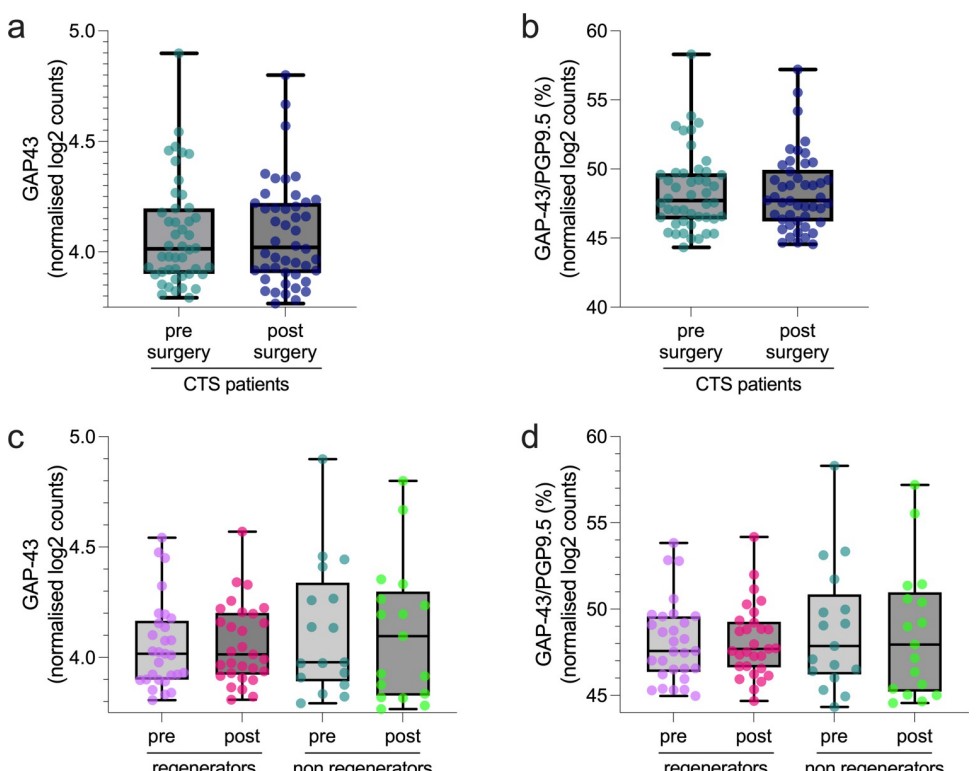

**Fig 3. GAP-43 Gene expression.** No difference in GAP-43 gene expression in skin of patients with CTS before and after carpal tunnel decompression. Normalised log2 counts of (A) Gap-43 and (B) GAP-43 gene expression expressed as a percentage of PGP9.5 gene expression are comparable before and after carpal tunnel surgery. No differences in (C) GAP-43 gene expression and (D) GAP-43 expressed as a percentage of PGP9.5 gene expression in skin of patients classified as regenerators and non-regenerators both before and after surgery.

## Discussion

Using CTS as a model system we demonstrated that patients have lower GAP-43+ IENFD before surgery than healthy controls however this difference vanished if correcting for total IENFD (proportions of PGP+ IENF containing GAP-43+). Surgery did not change GAP-43 expression in IENF, with no differences both in patients who were classified as regenerators and non-regenerators. Similarly, we could not identify differences in cutaneous GAP-43 gene expression before to after surgery and patients classified as regenerators or non-regenerators. This suggests that GAP-43 may not be a compelling marker for cutaneous human nerve regeneration.

Similar to reports from patients with systemic peripheral neuropathies [11–14], GAP-43 staining in cutaneous nerve fibres was downregulated in patients with focal nerve injury compared to healthy controls. However, when correcting for the total number of intraepidermal nerve fibres (percentage of PGP+ IENF containing GAP43), this difference disappeared. This suggests that this difference was driven by an overall reduction in intraepidermal nerve fibres rather than changes to GAP-43 alone. This is in line with previous reports, which did not find changes in the proportion of GAP-43+ IENF in patients with a range of peripheral neuropathies when correcting for overall nerve fibre density [14,15]. In contrast, other studies have demonstrated a decreased proportion of GAP-43+ IENFD in patients with diabetic neuropathy [11,12].

Of note, we did not find an association between the expression of pre and post surgical GAP-43 in IENF and regenerative capacity. Preclinical reports have associated GAP-43 with peripheral nerve regeneration, although most studies examined GAP-43 expression at the level of the dorsal root ganglia, spinal cord or site of nerve injury mid axon rather than in the skin [4–6, for review see 25]. More recently, it has been suggested that the role of GAP-43 in the coordination of axonal plasticity may not be pivotal but rather supporting, through coordination of other critical factors [26]. In humans, GAP-43 expression particularly in cutaneous axons has repeatedly been examined in the context of peripheral neuropathies and links with regenerative capacity were drawn. However, to date it has remained unclear whether cutaneous GAP-43 is indeed a valid marker for human nerve regeneration. The few longitudinal studies available either did not examine a direct relationship of GAP-43 expression and nerve regeneration [27], or did not find a temporal change in GAP-43 [13,15]. Narayanaswamy et al [13] examined changes in GAP-43 in cutaneous calf nerve fibres of patients with diabetic polyneuropathy over a six-month period. Whereas they found a continuing degeneration of PGP + IENF, only one out of 29 patients had GAP-43+ IENF and no differences in subepidermal GAP-43 staining were observed. Similarly, Albayrak et al [15] did not find a change in GAP-43 staining before and after chemotherapy and no association with epidermal nerve fibre density. This is in line with our findings that GAP-43 expression was not associated with regeneration or degeneration. The only longitudinal study to our knowledge that showed a change in GAP-43 was a small case report on two patients with chronic inflammatory demyelinating polyneuropathies. In this report, skin mRNA levels of GAP-43 were higher in patients than healthy controls and this normalised after 3 months of immunotherapy. Given the presence of GAP-43 in most IENFD even in healthy human skin [9,10], it is possible that the expression of this marker is representative of continuous remodelling of cutaneous afferents with natural skin turnover, as opposed to injury induced nerve regeneration per se.

Whilst our data suggests that GAP-43 in skin may not be a useful marker for human cutaneous nerve regeneration, several points need to be considered. First, our patients had long-standing CTS, with a mean duration of ~ 3 years. Previous work suggests that GAP-43 is changed particularly in patients with shorter disease duration; mRNA in the proximal thigh is

only elevated in patients with peripheral neuropathies and disease duration <3 years [14]. As such, we cannot exclude that GAP-43 may be a valid marker at earlier stages of neuropathy. Second, increased GAP-43 immunoreactivity in IENF has previously been reported particularly in patients with burning pain and those with allodynia [28]. None of our patients presented with allodynia and burning pain is not a predominant symptom in CTS according to the Neuropathic Pain Symptom Inventory (mean = 2.23, SD = 3.06, range 0–10). In accordance with a recent study in patients with chemotherapy induced neuropathy[15], we did not find a correlation between GAP-43 staining and symptoms in patients with CTS (S2 Table). Third, this is the first study examining GAP-43 expression in the glabrous finger skin in patients with peripheral neuropathy. Previous studies examined hairy leg biopsies. Whereas innervation densities differ between hairy and glabrous skin [29], epidermal sensory afferent regeneration capacity seems largely comparable in both types of skin [30]. Lastly, we re-examined the skin biopsies at 6 months after surgery. Potentially, changes in GAP-43 may only be detectable at early time points after decompression surgery and may return to normal levels at 6 months. Nevertheless, we have shown that pre-surgical GAP-43 was not associated with regenerative capacity.

In summary, using histological staining methods and RNA sequence analysis, we have found no association between neurocutaneous GAP-43 expression levels and regeneration in humans with chronic focal nerve injury. This questions the use of GAP-43 as a cutaneous marker for human nerve regeneration.

## Supporting information

**S1 Fig. No association between PGP+ IENF regeneration and the percentage of PGP+IENF containing GAP-43.** Graph shows pre surgical (green) and post-surgical (blue) PGP+ IENF containing GAP-43 with cubic spline curves showing no association.
(TIFF)

**S1 Table. Median nerve neurophysiology data of healthy participants and patients with CTS pre- and post-surgery.** Data are presented as median [interquartile range].
(DOCX)

**S2 Table. Spearman correlations between pre and post-operative cutaneous GAP-43 expression (histological analysis) and pain characteristics according to the Neuropathic Pain Symptom Inventory.** Table shows p-values. No significant correlations were identified.
(DOCX)

## Acknowledgments

We would like to thank all participants for volunteering in our study. The help of the hand surgeons at Oxford University Hospital Trust and the Nurses of the Clinical Research Network Thames Valley in the recruitment of participants is gratefully acknowledged.

## Author Contributions

**Conceptualization:** Annina Schmid.

**Data curation:** Liam Carroll, Oliver Sandy-Hindmarch, Georgios Baskozos, Annina Schmid.

**Formal analysis:** Liam Carroll, Georgios Baskozos, Guan Cheng Zhu, Julia McCarthy, Annina Schmid.

**Funding acquisition:** Annina Schmid.

**Investigation:** Annina Schmid.

**Methodology:** Liam Carroll, Oliver Sandy-Hindmarch, Georgios Baskozos, Guan Cheng Zhu, Julia McCarthy, Annina Schmid.

**Project administration:** Liam Carroll, Oliver Sandy-Hindmarch, Annina Schmid.

**Resources:** Georgios Baskozos.

**Supervision:** Annina Schmid.

**Visualization:** Georgios Baskozos, Annina Schmid.

**Writing – original draft:** Liam Carroll, Annina Schmid.

**Writing – review & editing:** Oliver Sandy-Hindmarch, Georgios Baskozos, Guan Cheng Zhu, Julia McCarthy, Annina Schmid.

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
