## [Decision Letter · Decision Letter 0]

22 Jun 2022

PONE-D-22-11884Cutaneous expression of Growth-Associated Protein 43 is not a compelling marker for human nerve regeneration after chronic focal nerve injuryPLOS ONE

Dear Dr. Schmid,

Thank you for submitting your manuscript to PLOS ONE. After careful consideration, we feel that it has merit but does not fully meet PLOS ONE’s publication criteria as it currently stands. Therefore, we invite you to submit a revised version of the manuscript that addresses the points raised by the reviewers.

We look forward to receiving your revised manuscript.

Kind regards,

Shao-Jun Tang

Academic Editor

PLOS ONE

Journal Requirements:

"ABS is supported by a Wellcome Trust Clinical Career Development Fellowship (222101/Z/20/Z) and the National Institute for Health Research (NIHR) Oxford Biomedical Research Centre (BRC). The views expressed are those of the authors and not necessarily those of the NHS, the NIHR or the Department of Health. The Oxford carpal tunnel cohort was supported by an advanced postdoc mobility fellowship from the Swiss National Science Foundation (P00P3-158835 to ABS) and an early career research grant from the International Association for the Study of Pain (to ABS). GB is supported by Diabetes UK (19/0005984). This research was funded in whole, or in part, by the Wellcome Trust [222101/Z/20/Z]. For the purpose of Open Access, the author has applied a CC BY public copyright license to any Author Accepted Manuscript version arising from this submission."

Reviewers' comments:

Reviewer's Responses to Questions

**Comments to the Author**

1. Is the manuscript technically sound, and do the data support the conclusions?

Reviewer #1: Yes

2. Has the statistical analysis been performed appropriately and rigorously? 

Reviewer #1: Yes

3. Have the authors made all data underlying the findings in their manuscript fully available?

Reviewer #1: Yes

4. Is the manuscript presented in an intelligible fashion and written in standard English?

Reviewer #1: Yes

5. Review Comments to the Author

Reviewer #1: Manuscript number: PONE-D-22-11884

Title: Cutaneous expression of Growth-Associated Protein 43 is not a compelling marker for human nerve regeneration after chronic focal nerve injury.

Overview and general recommendation:

GAP43 is a well-known marker for neuronal development/growth in embryo and regeneration after neuronal injury in adult rodent animal. Capral tunnel syndrome (CTS) is a typical model both for human sensory nerve degeneration due to nerve compression and regeneration after surgery decompression. The strength of the paper is that Carrol et al choose the optimized model to investigate the difference of GAP43+ immune-positive cutaneous nerve fibers before and after surgery and whether GAP43 expression is related to after surgery regeneration; The authors provided the meaningful and realistic results of GAP43+ fiber regeneration in chronic nerve injury of CTS in patients. This article is acceptable for publication after the. following points are addressed.

The major concerns:

Title: As this study only investigates the carpal tunnel syndrome patient’s index finger skin sample with average symptom duration 37.6 months, which belongs to chronic neuronal injury, it should be cautioned to expand the conclusion to all chronic focal nerve injury which may involve different mechanisms. I suggest that the title should be specified to carpal tunnel syndrome only.

The GAP43 immunofluorescence staining image: The study measures and expresses the GAP43+ fiber in a reasonable scientific way (Fig 1 and 2) with matched statistics calculation. I strongly suggest the authors provide data from PGP9.5 and GAP43 staining to show the audience that both fibers are effectively detected.

6. PLOS authors have the option to publish the peer review history of their article (what does this mean?). If published, this will include your full peer review and any attached files.

Reviewer #1: No

---

## [Author Response · Author response to Decision Letter 0]

29 Jul 2022

Point by point reply

1. Title: As this study only investigates the carpal tunnel syndrome patient’s index finger skin sample with average symptom duration 37.6 months, which belongs to chronic neuronal injury, it should be cautioned to expand the conclusion to all chronic focal nerve injury which may involve different mechanisms. I suggest that the title should be specified to carpal tunnel syndrome only. 

We have now changed out title as suggested to read: “Cutaneous expression of Growth-Associated Protein 43 is not a compelling marker for human nerve regeneration in carpal tunnel syndrome”

2. The GAP43 immunofluorescence staining image: The study measures and expresses the GAP43+ fiber in a reasonable scientific way (Fig 1 and 2) with matched statistics calculation. I strongly suggest the authors provide data from PGP9.5 and GAP43 staining to show the audience that both fibers are effectively detected.

We agree that the inclusion of staining images is helpful to assure the readers of the quality of our staining. We have now added a new Figure 2 including confocal immunohistochemistry images demonstrating staining of GAP-43 and PGP in skin sections. 

Editorial items

We have layouted our article according to the guidelines.

We have provided the updated funding statement in the cover letter.

We have updated the data availability statement and now provide both RNAseq and phenotypic/staining data in open access (see cover letter and manuscript).

We have adjusted the abstract in the manuscript. 

We have adjusted our supporting information files and citations in accordance with PLOSone guidelines.

---

## [Decision Letter · Decision Letter 1]

4 Oct 2022

PONE-D-22-11884R1Cutaneous expression of Growth-Associated Protein 43 is not a compelling marker for human nerve regeneration in carpal tunnel syndromePLOS ONE

Dear Dr. Schmid,

Thank you for submitting your manuscript to PLOS ONE. After careful consideration, we feel that it has merit but does not fully meet PLOS ONE’s publication criteria as it currently stands. Therefore, we invite you to submit a revised version of the manuscript that addresses the points raised during the review process. I agree with the reviewer that adding more images to Fig. 2 would be beneficial (especially those that show nerve terminals). 

Please submit your revised manuscript in 45 days after this decision letter. If you will need more time than this to complete your revisions, please reply to this message or contact the journal office at plosone@plos.org. Please include the following items when submitting your revised manuscript:A rebuttal letter that responds to each point raised by the academic editor and reviewer(s). You should upload this letter as a separate file labeled 'Response to Reviewers'.A marked-up copy of your manuscript that highlights changes made to the original version. You should upload this as a separate file labeled 'Revised Manuscript with Track Changes'.An unmarked version of your revised paper without tracked changes. You should upload this as a separate file labeled 'Manuscript'.If applicable, we recommend that you deposit your laboratory protocols in protocols.io to enhance the reproducibility of your results. Protocols.io assigns your protocol its own identifier (DOI) so that it can be cited independently in the future. For instructions see: https://journals.plos.org/plosone/s/submission-guidelines#loc-laboratory-protocols. Additionally, PLOS ONE offers an option for publishing peer-reviewed Lab Protocol articles, which describe protocols hosted on protocols.io. Read more information on sharing protocols at https://plos.org/protocols?utm_medium=editorial-email&utm_source=authorletters&utm_campaign=protocols.

We look forward to receiving your revised manuscript.

Kind regards,

Michal Hetman

Academic Editor

PLOS ONE

Journal Requirements:

Reviewers' comments:

Reviewer's Responses to Questions

**Comments to the Author**

1. If the authors have adequately addressed your comments raised in a previous round of review and you feel that this manuscript is now acceptable for publication, you may indicate that here to bypass the “Comments to the Author” section, enter your conflict of interest statement in the “Confidential to Editor” section, and submit your "Accept" recommendation.

Reviewer #1: All comments have been addressed

2. Is the manuscript technically sound, and do the data support the conclusions?

Reviewer #1: Partly

3. Has the statistical analysis been performed appropriately and rigorously? 

Reviewer #1: Yes

4. Have the authors made all data underlying the findings in their manuscript fully available?

Reviewer #1: Yes

5. Is the manuscript presented in an intelligible fashion and written in standard English?

Reviewer #1: Yes

6. Review Comments to the Author

Reviewer #1: 08.15.22 Resubmission review

1. Title. I really appreciate that the author changed the title to make it specific to carpal tunnel syndrome.

2. The GAP43 immunofluorescence staining image needs minor revise. In figure 2, the PGP9.5 staining appears correct, however the GAP43 staining presents minor problem. Most of the GAP43 Cy3 color overlaps with PGP9.5 (AlexaFluor-488) and fails to show the difference between the two nociceptors. Consideration It is true that most GAP43+ fibers are usually positive for PGP9.5, even they might be different nociceptors, derived from different types of DRG neuron, thus they should have their own unique morphology. The overlapped immunofluorescence staining is a weakness of this image. Another weakness of this figure is the nerve bundle traveling inside the dermis is well presented, however, the axon terminals located inside the epidermis which is the most important part to distinguish the different type of nociceptor are not well stained. Therefore, it is difficult to discern the two types of nociceptors. Overall, considering the weakness of figure 2, I would suggest the author provide additional PGP9.5 and GAP43 double staining image with the emphasis on the fiber located within epidermis as a minor revise.

7. PLOS authors have the option to publish the peer review history of their article (what does this mean?). If published, this will include your full peer review and any attached files.

Reviewer #1: No

---

## [Author Response · Author response to Decision Letter 1]

19 Oct 2022

Reviewer 1: The GAP43 immunofluorescence staining image needs minor revise. In figure 2, the PGP9.5 staining appears correct, however the GAP43 staining presents minor problem. Most of the GAP43 Cy3 color overlaps with PGP9.5 (AlexaFluor-488) and fails to show the difference between the two nociceptors. It is true that most GAP43+ fibers are usually positive for PGP9.5, even they might be different nociceptors, derived from different

types of DRG neuron, thus they should have their own unique morphology. The overlapped immunofluorescence staining is a weakness of this image. 

Another weakness of this figure is the nerve bundle traveling inside the dermis is well presented, however, the axon terminals located inside the epidermis which is the most important part to distinguish the different type of nociceptor are not well stained. Therefore, it is difficult to discern the two types of nociceptors. Overall, considering the weakness of figure 2, I would suggest the author provide additional PGP9.5 and GAP43 double staining image with the emphasis on the fiber located within epidermis as a minor revise.

Response: We would like to thank the reviewer for this comment. We have now taken new images of the double staining of PGP and GAP-43 and revised our figure accordingly. 

However, we are somewhat puzzled by the comment. PGP9.5 stains all nerve fibres, it is a pan-axonal stain. There are therefore no nerve fibres that are GAP-43 positive but PGP9.5 negative. As such GAP-43 does not stain a different subset of nociceptors than PGP9.5 would, but rather GAP-43+ neurons are a subset within PGP+ nerve fibres. We agree though that in the image we provided, most PGP+ fibres were also GAP-43 positive. That is because we selectively chose a section of a patient with overlap to show the staining quality. 

We have now revised our figure to include a representative image which shows some PGP+ intraepidermal nerve fibres with and some without GAP-43 labelling. We have highlighted these with arrows and arrow heads respectively.

We have now also delineated the dermal/epidermal border to help the readers distinguish between dermal and intraepidermal nerve fibres, as in fact our previous image contained many intraepidermal nerve fibres. Please note that the morphology of finger skin is quite distinct (e.g., many papillary indentations) from the leg. This may have led to the misinterpretation of the images to only contain dermal fibres. We hope that these revisions and the new images will help showcase the quality of our staining as well as the double labelling better.

---

## [Editor Report · Decision Letter 2]

21 Oct 2022

Cutaneous expression of Growth-Associated Protein 43 is not a compelling marker for human nerve regeneration in carpal tunnel syndrome

PONE-D-22-11884R2

Dear Dr. Schmid,

We’re pleased to inform you that your manuscript has been judged scientifically suitable for publication and will be formally accepted for publication once it meets all outstanding technical requirements.

Kind regards,

Michal Hetman

Academic Editor

PLOS ONE
---

## [Editor Report · Acceptance letter]

8 Nov 2022

PONE-D-22-11884R2 

Cutaneous expression of Growth-Associated Protein 43 is not a compelling marker for human nerve regeneration in carpal tunnel syndrome 

Dear Dr. Schmid:

I'm pleased to inform you that your manuscript has been deemed suitable for publication in PLOS ONE. Congratulations! Your manuscript is now with our production department. 

Kind regards, 

on behalf of

Dr. Michal Hetman 

Academic Editor

PLOS ONE